# Learning Regularized Monotone Graphon Mean-Field Games

**Fengzhuo Zhang**[1]    **Vincent Y. F. Tan**[1]    **Zhaoran Wang**[2]    **Zhuoran Yang**[3]

[1]National University of Singapore    [2] Northwestern University    [3]Yale University

fzzhang@u.nus.edu, vtan@nus.edu.sg,
zhaoranwang@gmail.com, zhuoranyang.work@gmail.com

## Abstract

This paper studies two fundamental problems in regularized Graphon Mean-Field Games (GMFGs). First, we establish the existence of a Nash Equilibrium (NE) of any $\lambda$-regularized GMFG (for $\lambda \geq 0$). This result relies on weaker conditions than those in previous works for analyzing both unregularized GMFGs ($\lambda = 0$) and $\lambda$-regularized MFGs, which are special cases of GMFGs. Second, we propose provably efficient algorithms to learn the NE in weakly monotone GMFGs, motivated by Lasry and Lions [2007]. Previous literature either only analyzed continuous-time algorithms or required extra conditions to analyze discrete-time algorithms. In contrast, we design a discrete-time algorithm and derive its convergence rate solely under weakly monotone conditions. Furthermore, we develop and analyze the action-value function estimation procedure during the online learning process, which is absent from algorithms for monotone GMFGs. This serves as a sub-module in our optimization algorithm. The efficiency of the designed algorithm is corroborated by empirical evaluations.

## 1 Introduction

In Multi-Agent Reinforcement Learning (MARL), the sizes of state and action spaces grow exponentially in the number of agents, which is known as the "curse of many agents" [Sonu et al., 2017, Wang et al., 2020] and restrict its applicability to large-scale scenarios. The Mean-Field Game (MFG) has thus been proposed to mitigate this problem by exploiting the homogeneity assumption [Huang et al., 2006, Lasry and Lions, 2007], and it has achieved tremendous successes in many real-world applications [Cousin et al., 2011, Achdou et al., 2020]. However, the homogeneity assumption is an impediment when modeling scenarios in which the agents are heterogeneous. GMFGs, as extensions of MFGs, have thus been proposed to model the behaviors of heterogeneous agents and ameliorate the "curse of many agents" at the same time [Parise and Ozdaglar, 2019, Carmona et al., 2022].

Despite the empirical successes of the Graphon Mean-Field Game (GMFG) [Aurell et al., 2022a], its theoretical understanding is lacking. First, sufficient conditions for Nash Equilibrium (NE) existence in regularized GMFG have not been established. Most works only address the existence of the NE in *unregularized* GMFGs. However, *regularization* is employed in practical implementations for improved exploration and robustness [Geist et al., 2019]. Moreover, previous works prove the existence of NE in regularized MFGs, a special case of GMFGs, only under the *contraction condition*, which is overly restrictive for real-world applications. Second, the analysis of *discrete-time* algorithms for monotone GMFGs is lacking. Most existing works design provably efficient discrete-time algorithms only under contraction conditions, as shown in Table 1. Complementarily, previous works on monotone GMFGs either only derive the convergence rate for continuous-time algorithms, which ignores the discretization error, or require extra conditions, (e.g., potential games) to analyze discrete-time algorithms.

37th Conference on Neural Information Processing Systems (NeurIPS 2023).

Table 1: Comparison of GMFGs and MFGs learning algorithms

| | Condition | No population manipulation | Online playing | Heterogeneity | Discrete-time algorithm | Convergence rate |
|---|---|---|---|---|---|---|
| Anahtarci et al. [2022] | Contraction | No | No | No | Yes | Yes |
| Xie et al. [2021a] | Contraction | No | Yes | No | Yes | Yes |
| Zaman et al. [2022] | Contraction | No | Yes | No | Yes | Yes |
| Yardim et al. [2022] | Contraction | Yes | Yes | No | Yes | Yes |
| Perrin et al. [2020] | Monotone | No | No | No | No | Yes |
| Geist et al. [2021] | Potential &Monotone | No | No | No | Yes | Yes |
| Perolat et al. [2021] | Monotone | Yes | Yes | Yes | No | No |
| Fabian et al. [2022] | Monotone | Yes | Yes | Yes | No | No |
| **Our work** | **Monotone** | **Yes** | **Yes** | **Yes** | **Yes** | **Yes** |

In this paper, we first consider GMFGs in full generality, i.e., without any contraction or monotone conditions. The goal is to establish the existence of NE in the regularized GMFG in this general setting. Then we focus on monotone GMFGs motivated by Lasry and Lions [2007]. We aim to learn the unique NE from the online interactions of all agents with and without the action-value function oracle. When the oracle is absent, the action-value functions should be estimated from the data of sampled agents generated in the online game.

In the analysis, we have to overcome difficulties that arise from both the *existence* of the NE problem and the *learning* of the NE. First, the proof of the existence of NE in regularized GMFG involves establishing some topological spaces and operators related to NE on which fixed point theorems are applicable. However, the direct construction of the space and the operators for GMFGs with uncountably infinite agents is challenging. Second, the design and analysis of the discrete-time NE learning algorithm require subtle exploitation of the monotone condition. Unlike continuous-time algorithms with infinitesimal step sizes, the design of appropriate step sizes is additionally required for the discrete-time algorithm to guarantee that iterates evolve appropriately. This guarantee originates from the delicate utilization of the monotone condition in the optimization procedures.

To address the difficulty of the existence problem, we construct a regularized MFG from the regularized GMFG and show that the NE of the constructed game can be converted into the NE of the original game, thus mitigating the difficulty of having an uncountable number of agents. To handle the difficulty in the NE learning problem, we design the Monotone GMFG Policy Mirror Descent (MonoGMFG-PMD) algorithm, which iteratively implements policy mirror descent for each player. We show that this procedure results in a decrease of the KL divergence between the iterate and the NE, and this decrease is related to the gap appearing in the weakly monotone condition. When the action-value function oracle is absent, we also design and analyze action-value functions estimation algorithms to serve as a submodule of the optimization procedures.

**Main Contributions:** We first establish the existence of the NE in the $\lambda$-regularized GMFG with $\lambda \geq 0$ assuming Lipschitzness of graphons and continuity of transition kernels and reward functions. Our result relaxes the assumption of the Lipschitzness of transition kernels and rewards required in previous works on unregularized GMFGs and the contraction condition in the literature on regularized MFG [Cui and Koeppl, 2021a]. Then we design and analyze the MonoGMFG-PMD algorithm. Using an action-value function oracle, the convergence rate for MonoGMFG-PMD is proved to be $\tilde{O}(T^{-1/2})$ after $T$ iterations. Without the oracle, the convergence rate includes an additional $\tilde{O}(K^{-1/2} + N^{-1})$ term that arises from sampling $N$ agents and collecting data from $K$ episodes, reflecting the generalization error and the approximation error of the estimation algorithm. As shown in Table 1 , our algorithm can be implemented from the online interaction of agents and does not require the distribution flow manipulation. Detailed explanations of the properties stated in the columns of Table 1 are provided in Appendix A. Our result for MonoGMFG-PMD provides the *first convergence rate for discrete-time algorithms in monotone GMFGs*.

## 2   Related Works

MFGs were proposed by Huang et al. [2006] and Lasry and Lions [2007] to model the interactions among a set of homogeneous agents. In recent years, learning the NE of the MFGs formulated by

discrete-time Markov Decision Process (MDP)s has attracted a lot of interest. There is a large body of works that design and analyze algorithms for the MFGs under contraction conditions [Anahtarci et al., 2019, 2022, Cui and Koeppl, 2021a, Xie et al., 2021a, Zaman et al., 2022, Yardim et al., 2022]. Typically, these works design reinforcement learning algorithms to approximate the contraction operators in MFGs, and the NE is learned by iteratively applying this operator. In contrast, another line of works focuses on the MFGs under monotone conditions. Motivated by Lasry and Lions [2007], the transition kernels in these works are independent of the mean fields. Perrin et al. [2020] proposed and analyzed the continuous-time fictitious play algorithm, which dynamically weighs the past mean fields and the best response policies to derive new mean fields and policies. With the additional potential assumption, Geist et al. [2021] derived the convergence rate for the discrete-time fictitious play algorithm. Perolat et al. [2021] then proposed the continuous-time policy mirror descent algorithm but only proved the asymptotic convergence, i.e., the consistency. In summary, there is no convergence rate result for any discrete-time algorithm for MFGs under the monotone condition. In addition, the relationship between the contraction conditions and the monotone conditions is not clear from existing works, but they complement each other.

To capture the potential heterogeneity among agents, GMFGs have been proposed by Parise and Ozdaglar [2019] in static settings as an extension of MFGs. The heterogeneous interactions among agents are represented by graphons. Aurell et al. [2022b], Caines and Huang [2019] then extended the GMFGs to the continuous-time setting, where the existence and the uniqueness of NE were established. Vasal et al. [2020] formulated discrete-time GMFGs and provided way to compute the NE with master equations. With the precise underlying graphons values, Cui and Koeppl [2021b] proposed algorithms to learn the NE of GMFGs with the contraction condition by modifying MFGs learning algorithms. Fabian et al. [2022] considered the monotone GMFG and proposed the continuous-time policy mirror descent algorithm to learn the NE. However, only consistency was provided in the latter two works.

**Notation** Let $[N] := \{1, \cdots, N\}$. Given a measurable space $(\Omega, \mathcal{F})$, we denote the collection of all the measures and the probability measures on $(\Omega, \mathcal{F})$ as $\mathcal{M}(\Omega)$ and $\Delta(\Omega)$, respectively. For a metric space $(\mathcal{X}, \| \cdot \|)$, we use $C(\mathcal{X})$ and $C_b(\mathcal{X})$ to denote the set of all continuous functions and the set of all bounded continuous functions on $\mathcal{X}$, respectively. For a measurable space $(\mathcal{X}, \mathcal{F})$ and two distributions $P, Q \in \Delta(\mathcal{X})$, the total variation distance between them is defined as $\mathrm{TV}(P, Q) := \sup_{A \in \mathcal{F}} |P(A) - Q(A)|$. A sequence of measures $\{\mu_n\}$ on $\mathcal{X}$ is said to *converge weakly* to a measure $\mu$ if $\int_{\mathcal{X}} g(x) \mu_n(\mathrm{d}x) \to \int_{\mathcal{X}} g(x) \mu(\mathrm{d}x)$ for all $g \in C_b(\mathcal{X})$.

## 3 Preliminaries

### 3.1 Graphon Mean-Field Games

We consider a GMFG that is defined through a tuple $(\mathcal{I}, \mathcal{S}, \mathcal{A}, H, P, r, W, \mu_1)$. The horizon (or length) of the game is denoted as $H$. In GMFG, we have infinite players, each corresponding to a point $\alpha \in \mathcal{I} = [0, 1]$. The state and action space of them are the same, denoted as $\mathcal{S} \subseteq \mathbb{R}^{d_s}$ and $\mathcal{A} \subseteq \mathbb{R}^{d_a}$ respectively. The interaction among players is captured by *graphons*. Graphons are symmetric functions that map $[0, 1]^2$ to $[0, 1]$. Symmetry here refers to that $W(\alpha, \beta) = W(\beta, \alpha)$ for all $\alpha, \beta \in [0, 1]$. We denote the set of graphons as $\mathcal{W} = \{W : [0, 1]^2 \to [0, 1] \,|\, W \text{ is symmetric.}\}$. The set of graphons of the game is $W = \{W_h\}_{h=1}^{H}$ with $W_h \in \mathcal{W}$. The state transition and reward of each player are influenced by the collective behavior of all the other players through an *aggregate* $z \in \mathcal{M}(\mathcal{S})$. At time $h \in [H]$, we denote the state distribution of player $\beta \in \mathcal{I}$ as $\mu_h^{\beta} \in \Delta(\mathcal{S})$. The aggregate for player $\alpha \in \mathcal{I}$ with the underlying graphon $W_h \in \mathcal{W}$ is then defined as

$$z_h^{\alpha} = \int_0^1 W_h(\alpha, \beta) \mu_h^{\beta} \, \mathrm{d}\beta. \tag{1}$$

The transition kernels $P = \{P_h\}_{h=1}^{H}$ of the game are functions $P_h : \mathcal{S} \times \mathcal{A} \times \mathcal{M}(\mathcal{S}) \to \mathcal{S}$ for all $h \in [H]$. At time $h$, if player $\alpha$ takes action $a_h^{\alpha} \in \mathcal{A}$ at state $s_h^{\alpha} \in \mathcal{S}$, her state will transition according to $s_{h+1}^{\alpha} \sim P_h(\cdot \,|\, s_h^{\alpha}, a_h^{\alpha}, z_h^{\alpha})$. The reward functions are denoted as $r_h : \mathcal{S} \times \mathcal{A} \times \mathcal{M}(\mathcal{S}) \to [0, 1]$ for all $h \in [H]$. We note that the players in GMFG are heterogeneous. This means that different players will, in general, receive different aggregates from other players. The distribution $\mu_1 \in \Delta(\mathcal{S})$ is the initial state distribution for all the players. A policy for an player $\alpha$ is $\pi^{\alpha} = \{\pi_h^{\alpha}\}_{h=1}^{H} \in \Pi^H$, where $\pi_h^{\alpha} : \mathcal{S} \to \Delta(\mathcal{A})$ takes action based only on the current state, and $\Pi$ is the set of all these policies.

A policy for all the players $\pi^{\mathcal{I}} \in \tilde{\Pi} = \Pi^{H \times \mathcal{I}}$ is the collection of the policies of each player, i.e, $\pi^{\mathcal{I}} = \{\pi^{\alpha}\}_{\alpha \in \mathcal{I}}$. In the following, we denote the state distributions of all the players at time $h$ and the state distributions of all the players at any time (distribution flow) respectively as $\mu_h^{\mathcal{I}} = \{\mu_h^{\alpha}\}_{\alpha \in \mathcal{I}}$ and $\mu^{\mathcal{I}} = \{\mu_h^{\mathcal{I}}\}_{h=1}^{H} \in \tilde{\Delta} = \Delta(\mathcal{S})^{H \times \mathcal{I}}$. Eqn. (1) shows that the aggregate $z_h^{\alpha}$ is a function of $\mu_h^{\mathcal{I}}$ and $W_h$, so to make this dependence explicit, we also write it as $z_h^{\alpha}(\mu_h^{\mathcal{I}}, W_h)$.

We consider the entropy-regularized GMFG. It has been shown that the regularization results in policy gradient algorithms converging faster [Shani et al., 2020, Cen et al., 2022]. In this game, the rewards of each player are the sum of the original rewards and the negative entropy of the policy multiplied by a constant. In a $\lambda$-regularized GMFG ($\lambda \geq 0$), the value function and the action-value function of player $\alpha$ with policy $\pi^{\alpha}$ on the MDP induced by the distribution flow $\mu^{\mathcal{I}}$ are defined as

$$V_h^{\lambda,\alpha}(s, \pi^{\alpha}, \mu^{\mathcal{I}}) = \mathbb{E}^{\pi^{\alpha}}\left[\sum_{t=h}^{H} r_t(s_t^{\alpha}, a_t^{\alpha}, z_t^{\alpha}(\mu_t^{\mathcal{I}}, W_t)) - \lambda \log \pi_t^{\alpha}(a_t^{\alpha} \mid s_t^{\alpha}) \,\middle|\, s_h^{\alpha} = s\right], \quad (2)$$

$$Q_h^{\lambda,\alpha}(s, a, \pi^{\alpha}, \mu^{\mathcal{I}}) = r_h(s, a, z_h^{\alpha}(\mu_h^{\mathcal{I}}, W_h)) + \mathbb{E}^{\pi^{\alpha}}\left[V_{h+1}^{\lambda,\alpha}(s_{h+1}^{\alpha}, \pi^{\alpha}, \mu^{\mathcal{I}}) \mid s_h^{\alpha} = s, a_h^{\alpha} = a\right] \quad (3)$$

for all $h \in [H]$, where the expectation $\mathbb{E}^{\pi^{\alpha}}$ is taken with respect to the stochastic process induced by implementing policy $\pi^{\alpha}$ on the MDP induced by $\mu^{\mathcal{I}}$. The cumulative reward function of player $\alpha$ is defined as $J^{\lambda,\alpha}(\pi^{\alpha}, \mu^{\mathcal{I}}) = \mathbb{E}_{\mu_1}[V_1^{\lambda,\alpha}(s, \pi^{\alpha}, \mu^{\mathcal{I}})]$. Then the notion of an NE is defined as follows.

**Definition 1.** *An NE of the $\lambda$-regularized GMFG is a pair $(\pi^{*,\mathcal{I}}, \mu^{*,\mathcal{I}}) \in \tilde{\Pi} \times \tilde{\Delta}$ that satisfies: (i) (player rationality) $J^{\lambda,\alpha}(\pi^{*,\alpha}, \mu^{*,\mathcal{I}}) = \max_{\tilde{\pi}^{\alpha} \in \Pi^H} J^{\lambda,\alpha}(\tilde{\pi}^{\alpha}, \mu^{*,\mathcal{I}})$ for all $\alpha \in \mathcal{I}$ up to a zero-measure set on $\mathcal{I}$ with respect to the Lebesgue measure. (ii) (Distribution consistency) The distribution flow $\mu^{*,\mathcal{I}}$ is equal to the distribution flow induced by implementing the policy $\pi^{*,\mathcal{I}}$.*

Similar to the NE for the finite-player games, the NE of the $\lambda$-regularized GMFG requires that the policy of each player is optimal. However, in GMFGs, the optimality is with respect to $\mu^{*,\mathcal{I}}$.

## 3.2  Mean-Field Games

As an important subclass of GMFG, MFG corresponds to the GMFG with *constant* graphons, i.e, $W(\alpha, \beta) = p$ for all $\alpha, \beta \in \mathcal{I}$. MFGs involve infinite *homogeneous* players. All the players employ the same policy and thus share the same distribution flow. The aggregate in Eqn. 1 degenerates to $z_h^{\alpha} = \int_0^1 p \cdot \mu_h^{\beta} \mathrm{d}\beta = p \cdot \mu_h$ for all $\alpha \in \mathcal{I}$. Here $\mu_h$ is the state distribution of a representative player. Thus, an MFG is denoted by a tuple $(\bar{\mathcal{S}}, \bar{\mathcal{A}}, \bar{H}, \bar{P}, \bar{r}, \mu_1)$. The state space, the action space, and the horizon are respectively denoted as $\bar{\mathcal{S}}$, $\bar{\mathcal{A}}$, and $\bar{H}$. Here, the transition kernels $\bar{P} = \{\bar{P}_h\}_{h=1}^{H}$ are functions $\bar{P}_h : \mathcal{S} \times \mathcal{A} \times \Delta(\mathcal{S}) \to \mathcal{S}$, and reward functions $\bar{r}_h : \mathcal{S} \times \mathcal{A} \times \Delta(\mathcal{S}) \to [0,1]$ for all $h \in [H]$. In the MFG, all the players adopt the *same* policy $\pi = \{\pi_h\}_{h=1}^{H}$ where $\pi_h \in \Pi$. The value function and the action-value function in the $\lambda$-regularized MFG with the underlying distribution flow $\mu = \{\mu_h\}_{h=1}^{H} \in \Delta(\mathcal{S})^H$ can be similarly defined as Eqn. (2) and (3) respectively as follows

$$\bar{V}_h^{\lambda}(s, \pi, \mu) = \mathbb{E}^{\pi}\left[\sum_{t=h}^{H} \bar{r}_t(s_t, a_t, \mu_t) - \lambda \log \pi_t(a_t \mid s_t) \,\middle|\, s_h = s\right],$$

$$\bar{Q}_h^{\lambda}(s, a, \pi, \mu) = \bar{r}_h(s, a, \mu_h) + \mathbb{E}^{\pi}\left[\bar{V}_{h+1}^{\lambda}(s_{h+1}, \pi, \mu) \mid s_h = s, a_h = a\right]$$

for all $h \in [H]$. The cumulative reward is $\bar{J}^{\lambda}(\pi, \mu) = \mathbb{E}_{\mu_1}[\bar{V}_1^{\lambda}(s, \pi, \mu)]$. The notion of NE can be similarly derived as follows.

**Definition 2.** *An NE of the $\lambda$-regularized MFG is a pair $(\pi^*, \mu^*) \in \Pi^H \times \Delta(\mathcal{S})^H$ that satisfies: (i) (player rationality) $J^{\lambda}(\pi^*, \mu^*) = \max_{\tilde{\pi} \in \Pi^H} J^{\lambda}(\tilde{\pi}, \mu^*)$. (ii) (Distribution consistency) The distribution flow $\mu^*$ is equal to the distribution flow induced by the policy $\pi^*$.*

**Remark 1.** *Compared with Definition 1, the definition of NE in MFG only involves the policy and the distribution flow of a single representative player, since the agents are homogeneous in MFGs.*

## 4  Existence of the NEs in Regularized GMFGs and MFGs

We now state some assumptions to demonstrate the existence of a NE for $\lambda$-regularized GMFGs.

**Assumption 1.** *The state space $\mathcal{S}$ is compact, and the action space $\mathcal{A}$ is finite.*

This assumption imposes rather mild constraints on $\mathcal{S}$ and $\mathcal{A}$. In real-world applications, the states are usually physical quantities and thus reside in a compact set. For the action space, many deep reinforcement learning algorithms discretize the potential continuous action sets into finite sets [Lowe et al., 2017, Mordatch and Abbeel, 2018].

**Assumption 2.** *The graphons $W_h$ for $h \in [H]$ are continuous functions.*

The stronger version of this assumption (Lipschitz continuity) is widely adopted in GMFG works [Cui and Koeppl, 2021b, Fabian et al., 2022]. It helps us to build the continuity of the transition kernels and rewards with respect to players.

**Assumption 3.** *For all $h \in [H]$, the reward function $r_h(s, a, z)$ is continuous on $\mathcal{S} \times \mathcal{A} \times \mathcal{M}(\mathcal{S})$, that is if $(s_n, a_n, z_n) \to (s, a, z)$ as $n \to \infty$, then $r_h(s_n, a_n, z_n) \to r_h(s, a, z)$. The transition kernel $P_h(\cdot \,|\, s, a, z)$ is weakly continuous in $\mathcal{S} \times \mathcal{A} \times \mathcal{M}(\mathcal{S})$, that is if $(s_n, a_n, z_n) \to (s, a, z)$ as $n \to \infty$, $P_h(\cdot \,|\, s_n, a_n, z_n) \to P_h(\cdot \,|\, s, a, z)$ weakly.*

This assumption states the continuity of the models, i.e., the transition kernels and the rewards, as functions of the state, action, and aggregate. We note that the Lipschitz continuity assumption of the model in the previous works implies that our assumption is satisfied [Cui and Koeppl, 2021b, Fabian et al., 2022]. Next, we state the existence result of regularized GMFG.

**Theorem 1.** *Under Assumptions 1, 2 and 3, for all $\lambda \geq 0$, the $\lambda$-regularized GMFG $(\mathcal{I}, \mathcal{S}, \mathcal{A}, H, P, r, W, \mu_1)$ admits an NE $(\pi^{\mathcal{I}}, \mu^{\mathcal{I}}) \in \tilde{\Pi} \times \tilde{\Delta}$.*

This theorem strengthens previous existence results in Cui and Koeppl [2021b] and Fabian et al. [2022] in two aspects. First, our assumptions are weaker. These two existing works require a finite state space and the Lipschitz continuity of the models. In contrast, we only need a compact state space and the model continuity. Second, their results only hold for the unregularized case ($\lambda = 0$), whereas ours holds for any $\lambda \geq 0$. In the proof of Theorem 1, we construct a MFG from the given GMFG and show that an NE of the constructed MFG can be converted to an NE of the GMFG. Then we prove the existence of NE in the constructed regularized MFG. Our existence result for the regularized MFG is also a significant addition to the MFG literature.

**Remark 2.** *Although we show that an NE of the constructed MFG can be converted to an NE of GMFG, this proof does not imply that GMFG forms a subclass of or is equivalent to MFG. This is because we have only demonstrated the relationship between the NEs of these two games, but the exact realizations of the GMFG and the conceptually constructed MFG may differ. It means that the sample paths of these two games may not be the same, which include the realizations of the states, actions, and rewards of all the players.*

We next state the assumption needed for the MFG.

**Assumption 4.** *The MFG $(\bar{\mathcal{S}}, \bar{\mathcal{A}}, \bar{H}, \bar{P}, \bar{r}, \mu_1)$ satisfies that: (i) The state space $\bar{\mathcal{S}}$ is compact, and the action space $\bar{\mathcal{A}}$ is finite. (ii) The reward functions $\bar{r}_h(s, a, \mu)$ for $h \in [H]$ are continuous on $\mathcal{S} \times \mathcal{A} \times \Delta(\mathcal{S})$. The transition kernels are weakly continuous on $\mathcal{S} \times \mathcal{A} \times \Delta(\mathcal{S})$; that is if $(s_n, a_n, \mu_n) \to (s, a, \mu)$ as $n \to \infty$, $\bar{P}_h(\cdot \,|\, s_n, a_n, \mu_n) \to \bar{P}_h(\cdot \,|\, s, a, \mu)$ weakly.*

Then the existence of the NE is stated as follows.

**Theorem 2.** *Under Assumption 4, the $\lambda$-regularzied MFG $(\bar{\mathcal{S}}, \bar{\mathcal{A}}, \bar{H}, \bar{P}, \bar{r}, \mu_1)$ admits an NE $(\pi, \mu) \in \Pi^H \times \Delta(\mathcal{S})^H$ for any $\lambda \geq 0$.*

Our result in Theorem 2 imposes weaker conditions than previous works [Cui and Koeppl, 2021a, Anahtarci et al., 2022] to guarantee the existence of an NE. These existing works prove the existence of NE by assuming a *contractive* property and the finiteness of the state space. They also require a strong Lispchitz assumption [Anahtarci et al., 2022], where the Lipschitz constants of the models should be small enough. In contrast, we only require the continuity assumption in Theorem 2. This is due to our analysis of the operator for the regularized MFG and the use of Kakutani fixed point theorem Guide [2006].

# 5 Learning NE of Monotone GMFGs

In this section, we focus on GMFGs with transition kernels that are independent of the aggregate $z$, i.e., $P_h : \mathcal{S} \times \mathcal{A} \to \mathcal{S}$ for $h \in [H]$. This model is motivated by the seminal work Lasry and Lions [2007], where the state evolution in continuous-time MFG is characterized by the Fokker–Plank equation. However, the form of the Fokker–Plank equation results in the state transition of each player being independent of other players. This model is also widely accepted in the discrete-time GMFG and MFG literature [Fabian et al., 2022, Perrin et al., 2020, Perolat et al., 2021].

## 5.1 Monotone GMFG

We first generalize the notion of *monotonicity* from multi-population MFGs in Perolat et al. [2021] to GMFGs.

**Definition 3** (Weakly Monotone Condition). *A GMFG is said to be* weakly monotone *if for any* $\rho^{\mathcal{I}}, \tilde{\rho}^{\mathcal{I}} \in \Delta(\mathcal{S} \times \mathcal{A})^{\mathcal{I}}$ *and their marginalizations on the states* $\mu^{\mathcal{I}}, \tilde{\mu}^{\mathcal{I}} \in \Delta(\mathcal{S})^{\mathcal{I}}$, *we have*

$$\int_{\mathcal{I}} \sum_{a \in \mathcal{A}} \int_{\mathcal{S}} \left( \rho^{\alpha}(s,a) - \tilde{\rho}^{\alpha}(s,a) \right) \left( r_h\big(s, a, z_h^{\alpha}(\mu^{\mathcal{I}}, W_h)\big) - r_h\big(s, a, z_h^{\alpha}(\tilde{\mu}^{\mathcal{I}}, W_h)\big) \right) \mathrm{d}s \, \mathrm{d}\alpha \le 0$$

*for all* $h \in [H]$, *where* $W_h$ *is the underlying graphon. It is* strictly weakly monotone *if the inequality is strict when* $\rho^{\mathcal{I}} \ne \tilde{\rho}^{\mathcal{I}}$.

In two MDPs induced by the distribution flows $\mu^{\mathcal{I}}$ of $\pi^{\mathcal{I}}$ and $\tilde{\mu}^{\mathcal{I}}$ of $\tilde{\pi}^{\mathcal{I}}$, the weakly monotone condition states that we can achieve higher rewards at stage $h \in [H]$ by swapping the policies. This condition has two important implications. The first is the uniqueness of the NE.

**Proposition 1.** *Under Assumptions 1, 2, and 3, a strictly weakly monotone $\lambda$-regularized GMFG has a unique NE for any $\lambda \ge 0$ up to a zero-measure set on $\mathcal{I}$ with respect to the Lebesgue measure.*

In the following, we denote this unique NE as $(\pi^{*,\mathcal{I}}, \mu^{*,\mathcal{I}})$, and we aim to learn this NE. The second implication concerns the relationship between the cumulative rewards of two policies.

**Proposition 2.** *If a $\lambda$-regularized GMFG satisfies the weakly monotone condition, then for any two policies $\pi^{\mathcal{I}}$, $\tilde{\pi}^{\mathcal{I}} \in \tilde{\Pi}$ and their induced distribution flows $\mu^{\mathcal{I}}, \tilde{\mu}^{\mathcal{I}} \in \tilde{\Delta}$, we have*

$$\int_0^1 J^{\lambda,\alpha}(\pi^{\alpha}, \mu^{\mathcal{I}}) + J^{\lambda,\alpha}(\tilde{\pi}^{\alpha}, \tilde{\mu}^{\mathcal{I}}) - J^{\lambda,\alpha}(\tilde{\pi}^{\alpha}, \mu^{\mathcal{I}}) - J^{\lambda,\alpha}(\pi^{\alpha}, \tilde{\mu}^{\mathcal{I}}) \, \mathrm{d}\alpha \le 0$$

*If the $\lambda$-regularized GMFG satisfies the strictly weakly monotone condition, then the inequality is strict when $\pi^{\mathcal{I}} \ne \tilde{\pi}^{\mathcal{I}}$.*

Proposition 2 shows that if we have two policies, we can improve the cumulative rewards on the MDP induced by these policies by swapping the policies. This implies an important property of the NE $(\pi^{*,\mathcal{I}}, \mu^{*,\mathcal{I}})$. Since $\pi^{*,\mathcal{I}}$ is optimal on the MDP induced by $\mu^{*,\mathcal{I}}$, we have $\int_0^1 J^{\lambda,\alpha}(\pi^{*,\alpha}, \mu^{*,\mathcal{I}}) \, \mathrm{d}\alpha \ge \int_0^1 J^{\lambda,\alpha}(\pi^{\alpha}, \mu^{*,\mathcal{I}}) \, \mathrm{d}\alpha$ for any $\pi^{\mathcal{I}} \in \tilde{\Pi}$. Then Proposition 2 shows that $\int_0^1 J^{\lambda,\alpha}(\pi^{*,\alpha}, \mu^{\mathcal{I}}) \, \mathrm{d}\alpha \ge \int_0^1 J^{\lambda,\alpha}(\pi^{\alpha}, \mu^{\mathcal{I}}) \, \mathrm{d}\alpha$ for any policy $\pi^{\mathcal{I}}$ and the distribution flow $\mu^{\mathcal{I}}$ it induces. This means that the NE policy gains cumulative rewards not less than any policy $\pi^{\mathcal{I}}$ on the MDP induced by $\pi^{\mathcal{I}}$. This motivates the design of our NE learning algorithm.

## 5.2 Policy Mirror Descent Algorithm for Monotone GMFG

In this section, we introduce the algorithm to learn the NE, which is called MonoGMFG-PMD and whose pseudo-code is outlined in Algorithm 1. It consists of three main steps. The first step estimates the action-value function (Line 3). We need to evaluate the action-value function of a policy on the MDP induced by itself. This estimate can be obtained for each player independently by playing the $\pi_t^{\mathcal{I}}$ several times. We assume access to a sub-module for this and quantify the estimation error in our analysis. The second step is the policy mirror descent (Line 4). Given $\lambda \eta_t < 1$, This step can be equivalently formulated as

$$\hat{\pi}_{t+1,h}^{\alpha}(\cdot \mid s) = \operatorname*{argmax}_{p \in \Delta(\mathcal{A})} \frac{\eta_t}{1 - \lambda \eta_t} \Big[ \langle \hat{Q}_h^{\lambda,\alpha}(s, \cdot, \pi_t^{\alpha}, \mu_t^{\mathcal{I}}), p \rangle - \lambda R(p) \Big] - D_{\mathrm{kl}}\big( p \| \pi_{t,h}^{\alpha}(\cdot \mid s) \big) \quad \forall s \in \mathcal{S},$$

---

**Algorithm 1** MonoGMFG-PMD

**Procedure:**
1: Initialize $\pi_{1,h}^{\alpha}(\cdot \mid s) = \mathrm{Unif}(\mathcal{A})$ for all $s \in \mathcal{S}, h \in [H]$ and $\alpha \in \mathcal{I}$.
2: **for** $t = 1, 2, \cdots, T$ **do**
3:  Compute the action-value function $\hat{Q}_h^{\lambda,\alpha}(s, a, \pi_t^{\alpha}, \mu_t^{\mathcal{I}})$ for all $\alpha \in \mathcal{I}$ and $h \in [H]$, where $\mu_t^{\mathcal{I}}$ is the distribution flow induced by $\pi_t^{\mathcal{I}}$.
4:  $\hat{\pi}_{t+1,h}^{\alpha}(\cdot \mid s) \propto \left(\pi_{t,h}^{\alpha}(\cdot \mid s)\right)^{1-\lambda\eta_t} \exp\left(\eta_t \hat{Q}_h^{\lambda,\alpha}(s, a, \pi_t^{\alpha}, \mu_t^{\mathcal{I}})\right)$ for all $\alpha \in \mathcal{I}$ and $h \in [H]$
5:  $\pi_{t+1,h}^{\alpha}(\cdot \mid s) = (1 - \beta_t)\hat{\pi}_{t+1,h}^{\alpha}(\cdot \mid s) + \beta_t \mathrm{Unif}(\mathcal{A})$
6: **end for**
7: Output $\bar{\pi}^{\mathcal{I}} = \mathrm{Unif}\left(\pi_{[1:T]}^{\mathcal{I}}\right)$

---

where $R(p) = \langle p, \log p \rangle$ is the negative entropy function. This step aims to improve the performance of the policy $\pi_t^{\mathcal{I}}$ on its own induced MDP. Intuitively, since the policy $\pi^{*,\mathcal{I}}$ in NE performs better than $\pi_t^{\mathcal{I}}$ on the MDP induced by $\mu_t^{\mathcal{I}}$ as shown in Section 5.1, the improved policy $\pi_{t+1}^{\mathcal{I}}$ should be closer to $\pi^{*,\mathcal{I}}$ than $\pi_t^{\mathcal{I}}$. The third step mixes the current policy with the uniform policy (Line 5) to encourage exploration.

MonoGMFG-PMD is different from previous NE learning algorithms for monotone GMFG in Perolat et al. [2021], Fabian et al. [2022] in three different ways. First, MonoGMFG-PMD is designed to learn the NE of the $\lambda$-regularized GMFG with $\lambda > 0$, whereas other algorithms learn the NE of the *unregularized* GMFGs. As a result, our policy improvement (Line 4) discounts the previous policy as $(\pi_{t,h}^{\alpha})^{1-\lambda\eta_t}$, but other algorithms retain $\pi_{t,h}^{\alpha}$. Second, our algorithm is discrete-time and thus amenable for practical implementation. However, other provably efficient algorithms evolve in continuous time. Finally, MonoGMFG-PMD encourages exploration in Line 5, which is important for the theoretical analysis. Such a step is missing in other algorithms.

### 5.3 Theoretical Analysis for MonoGMFG-PMD with Estimation Oracle

This section provides theoretical analysis for the MonoGMFG-PMD algorithm given an action-value function oracle in Line 3, i.e., $\hat{Q}_h^{\lambda,\alpha} = Q_h^{\lambda,\alpha}$. We denote the unique NE of the $\lambda$-regularized GMFG as $(\pi^{*,\mathcal{I}}, \mu^{*,\mathcal{I}})$. For any policy $\pi^{\mathcal{I}}$, we measure the distance to the policy $\pi^{*,\mathcal{I}}$ of NE using

$$D(\pi^{\mathcal{I}}) = \int_0^1 \sum_{h=1}^{H} \mathbb{E}_{\mu_h^{*,\alpha}}\left[D_{\mathrm{kl}}\left(\pi_h^{*,\alpha}(\cdot \mid s_h^{\alpha}) \| \pi_h^{\alpha}(\cdot \mid s_h^{\alpha})\right)\right] \mathrm{d}\alpha.$$

This metric measures the weighted KL divergence between policy $\pi^{\mathcal{I}}$ and the NE policy, and the weights are the NE distribution flow $\mu^{*,\mathcal{I}}$.

**Theorem 3.** *Assume that the GMFG is strictly weakly monotone and we have an action-value function oracle. Let $\eta_t = \eta = O(T^{-1/2})$ and $\beta_t = \beta = O(T^{-1})$ in MonoGMFG-PMD. For any $\lambda > 0$ we have*

$$D\left(\frac{1}{T} \sum_{t=1}^{T} \pi_t^{\mathcal{I}}\right) = O\left(\frac{\lambda \log^2 T}{T^{1/2}}\right).$$

Theorem 3 provides the first convergence rate result for a discrete-time algorithm on strictly weakly monotone GMFGs under mild assumptions. In contrast, Perolat et al. [2021], Fabian et al. [2022] only consider the continuous-time policy mirror descent, which is difficult for the practical implementation, and only provide the asymptotic consistency results. Geist et al. [2021] derive exploitability results for a fictitious play algorithm but require the potential structure and the Lipschitzness of the reward function. Our proof for Theorem 3 mainly exploits properties of NE discussed in Section 5.1. Concretely, we use the fact that the policy mirror descent procedure reduces the distance between the policy iterate and the NE as $D(\pi_{t+1}^{\mathcal{I}}) - D(\pi_t^{\mathcal{I}}) \approx \int_0^1 J^{\lambda,\alpha}(\pi_t^{\alpha}, \mu_t^{\mathcal{I}}) - J^{\lambda,\alpha}(\pi^{*,\alpha}, \mu_t^{\mathcal{I}}) \mathrm{d}\alpha$. Thus, the policy iterate becomes closer to the NE. However, the discretization error and the exploration influence (Line 5) also appear, requiring additional care to show that $D(\pi_t^{\mathcal{I}})$, in general, decreases.

---

**Algorithm 2** Estimation of Action-value Function

**Procedure:**
 1: Sample $N$ players $\{i/N\}_{i=1}^N \subseteq [0, 1]$.
 2: The $i^{\text{th}}$ player implements $\pi_t^{\text{b},i}$ for $i \in [N]$, and the other players implement $\pi_t^{\mathcal{I}}$.
 3: Collect data $\{(s_{\tau,h}^i, a_{\tau,h}^i, r_{\tau,h}^i)\}_{i,\tau,h=1}^{N,K,H}$ of sampled players from $K$ independent episodes.
 4: Initialize $\hat{V}_{H+1}^{\lambda,i}(s, a) = 0$ for all $s \in \mathcal{S}$, $a \in \mathcal{A}$ and $i \in [N]$.
 5: **for** time $h = H, \cdots, 1$ **do**
 6:   **for** Player $i = 1, \cdots, N$(in parallel) **do**
 7:     $\hat{Q}_h^{\lambda,i} = \operatorname{argmin}_{f \in \mathcal{F}_h} \sum_{\tau=1}^K \left( f(s_{\tau,h}^i, a_{\tau,h}^i) - r_{\tau,h}^i - \hat{V}_{h+1}^{\lambda,i}(s_{\tau,h+1}^i) \right)^2$.
 8:     $\hat{V}_h^{\lambda,i}(s) = \langle \hat{Q}_h^{\lambda,i}(s, \cdot), \pi_{t,h}^{i/N}(\cdot, s) \rangle - \lambda R\big(\pi_{t,h}^{i/N}(\cdot, s)\big)$.
 9:   **end for**
10: **end for**
11: Output $\{\hat{Q}_h^{\lambda,i}\}_{i,h=1}^{N,H}$.

---

## 5.4 Theoretical Analysis for MonoGMFG-PMD with General Function Classes

In this section, we remove the requirement that one is given oracle access to an action-value function and we estimate it in Line 3 of MonoGMFG-PMD using general function classes. We consider the action-value function class $\mathcal{F} = \mathcal{F}_1 \times \cdots \times \mathcal{F}_H$, where $\mathcal{F}_h \subseteq \{f : \mathcal{S} \times \mathcal{A} \to [0, (H - h + 1)(1 + \lambda \log |\mathcal{A}|)]\}$ is the class of action-value functions at time $h \in [H]$. Then we estimate the action-value functions using Algorithm 2.

This algorithm mainly involves two steps. The first is involves data collection (Line 3). Here we assign policies to players and collect data from their interactions. We let the sampled $N$ players implement behavior policies $\{\pi_t^{\text{b},i}\}_{i=1}^N$, which can be different from $\{\pi_t^{i/N}\}_{i=1}^N$. This will not change the aggregate $z_h^\alpha(\mu_h^{\mathcal{I}}, W_h)$ for any $\alpha \in \mathcal{I}$, since only a zero-measure set of players change their policies. The second is the action-value function estimation (Lines 7 and 8). The action-value function is selected based on the previous value function, and the value function is updated from the derived estimate. This can be implemented in parallel for all players. We highlight that the estimation analysis cannot leverage results from general non-parametric regression [Wainwright, 2019], since the response variable $\hat{V}_{h+1}^{\lambda,i}$ is *not* independent of $s_{\tau,h+1}^i$ in our setting.

**Assumption 5** (Realizability). *For any policy $\pi^{\mathcal{I}} \in \tilde{\Pi}$ and the induced distribution flow $\mu^{\mathcal{I}} \in \tilde{\Delta}$, we have $Q_h^{\lambda,\alpha}(\cdot, \cdot, \pi^\alpha, \mu^{\mathcal{I}}) \in \mathcal{F}_h$ for $h \in [H]$.*

This assumption ensures that we can find the nominal action-value function in the function class. For a policy $\pi \in \Pi^H$ and a function $f : \mathcal{S} \times \mathcal{A} \to \mathbb{R}$, we define the operator $(\mathcal{T}_h^\pi f)(s, a) = \mathbb{E}_{s' \sim P_h(\cdot|s,a)}[\langle f(s', \cdot), \pi_{h+1}(\cdot|s') \rangle - \lambda R(\pi_{h+1}(\cdot|s'))]$. For a policy $\pi^{\mathcal{I}}$ and the induced distribution flow $\mu^{\mathcal{I}}$, we have $Q_h^{\lambda,\alpha}(s, a, \pi^\alpha, \mu^{\mathcal{I}}) = r_h(s, a, z_h^\alpha) + (\mathcal{T}_h^{\pi^\alpha} Q_{h+1}^{\lambda,\alpha})(s, a)$.

**Assumption 6** (Completeness). *For any policy $\pi^{\mathcal{I}} \in \tilde{\Pi}$ and the induced distribution flow $\mu^{\mathcal{I}} \in \tilde{\Delta}$, we have that for all $f \in \mathcal{F}_{h+1}$, $r_h(\cdot, \cdot, z_h^\alpha(\mu_h^{\mathcal{I}}, W_h)) + (\mathcal{T}_h^{\pi^\alpha} f) \in \mathcal{F}_h$ for all $\alpha \in \mathcal{I}$, $h \in [H - 1]$.*

This completeness assumption ensures that the estimates from $\mathcal{F}$ also satisfy the relationship between nominal action-value functions through $\mathcal{T}_h^{\pi^\alpha}$. These realizability and completeness assumptions are widely adopted in the off-policy evaluation and offline reinforcement learning literature [Uehara et al., 2022, Xie et al., 2021b].

**Assumption 7.** *The reward functions $\{r_h\}_{h=1}^H$ are Lipschitz in $z$, i.e., $|r_h(s, a, z) - r_h(s, a, z')| \leq L_r \|z - z'\|_1$ for all $s \in \mathcal{S}, a \in \mathcal{A}, h \in [H]$. The graphons $W_h$ for $h \in [H]$ are Lipschitz continuous functions, i.e., there exists a constant $L = L_W > 0$ (depending only on $W = \{W_h\}_{h=1}^H$) such that $|W_h(\alpha, \beta) - W_h(\alpha', \beta')| \leq L_W(|\alpha - \alpha'| + |\beta - \beta'|)$ for all $\alpha, \alpha', \beta, \beta' \in [0, 1]$, and $h \in [H]$.*

The Lipschitzness assumption is common in the GMFG works [Parise and Ozdaglar, 2019, Carmona et al., 2022, Cui and Koeppl, 2021b]. It helps us to approximate the action-value function of a player by that of sampled players. We denote the state distributions of player $i$ induced by policy $\pi_t^{\text{b},i}$ as $\mu_t^{\text{b},i}$. Then we require the behavior policies $\{\pi_t^{\text{b},i}\}_{i=1}^N$ to satisfy the following requirements.

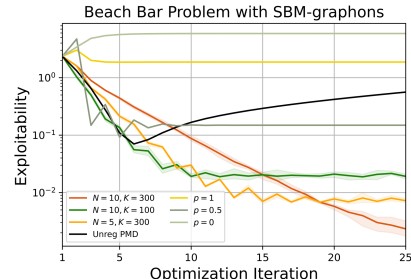
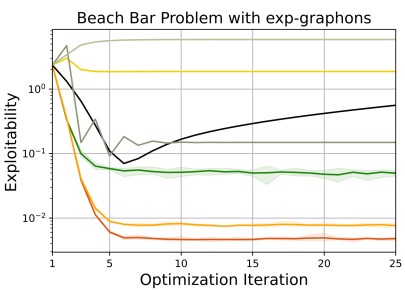

(a) Beach Bar problem with SBM graphons

(b) Beach Bar problem with exp-graphons.

Figure 1: Simulation results for Beach Bar problem with SBM and exp-graphons.

**Assumption 8.** *For any $t \in [T]$, the behavior policies explore sufficiently. More precisely, for any policy $\pi \in \Pi^H$ and induced distributions $\mu \in \Delta(\mathcal{S})^H$, we have $\sup_{s \in \mathcal{S}, a \in \mathcal{A}} \pi_h(a \mid s)/\pi_{t,h}^{b,i}(a \mid s) \leq C_1$ and $\sup_{s \in \mathcal{S}} \mathrm{d}\mu_h(s)/\mathrm{d}\mu_{t,h}^{b,i}(s) \leq C_2$ for $h \in [H]$ and $i \in [N]$ where $C_1, C_2 > 0$ are constants.*

This assumption guarantees that the behavior policies explore the actions that may be adopted by $\pi_t^{\mathcal{I}}$ and $\pi^{*,\mathcal{I}}$. Such an assumption is widely adopted in offline reinforcement learning and off-policy evaluation works [Uehara et al., 2022, Xie et al., 2021b].

**Theorem 4.** *Assume that the GMFG is weakly monotone and that Assumptions 5, 6, 7, and 8 hold. Let $\eta_t = \eta = O(T^{-1/2})$ and $\beta_t = \beta = O(T^{-1})$ in Algorithm 1 (MonoGMFG-PMD). Then with probability at least $1 - \delta$, Algorithms 1 and 2 yield*

$$D\left(\frac{1}{T}\sum_{t=1}^{T} \pi_t^{\mathcal{I}}\right) = O\left(\frac{\lambda \log^2 T}{\sqrt{T}} + C_1 C_2 \frac{H^{3/2} B_H^2}{\lambda \sqrt{K}} \log \frac{TNH \cdot \mathcal{N}_\infty(5B_H/K, \mathcal{F}_{[H]})}{\delta} + \frac{H \log T}{N}\right),$$

*where $B_H = H(1 + \lambda \log |\mathcal{A}|)$, and $\mathcal{N}_\infty(5B_H/K, \mathcal{F}_{[H]}) = \max_{h \in [H]} \mathcal{N}_\infty(5B_H/K, \mathcal{F}_h)$ is the $\ell_\infty$-covering number of the function class.*

The error in Theorem 4 consists of both the optimization and estimation errors. The optimization error corresponds to the first term, which also appears in Theorem 3. The estimation error consists of the generalization error and the approximation error, in the second and third terms respectively. When the function class $\mathcal{F}$ is finite, this term scales as $O(K^{-1/2})$, which originates from the fact that we estimate the action-value function from the empirical error instead of its population counterpart. The approximation error scales as $O(N^{-1})$. This term originates from the fact that the action-value function of player $\alpha$ is approximated by that of the sampled player near $\alpha$. To learn a policy that is at most $\varepsilon > 0$ far from the NE, we can set $T = \tilde{O}(\varepsilon^{-2})$, $K = \tilde{O}(\varepsilon^{-2})$, and $N = \tilde{O}(\varepsilon^{-1})$, which in total results in $TK = \tilde{O}(\varepsilon^{-4})$ episodes of online plays.

## 6 Experiments

In this section, we conduct experiments to corroborate our theoretical findings. We run different algorithms on the Beach Bar problem Perrin et al. [2020], Fabian et al. [2022]. The underlying graphons are set to Stochastic Block Model (SBM) and exp-graphons. The details of experiments are deferred to Appendix B. Since the NEs of the games are not available, we adopt the *exploitability* to measure the proximity between a policy and the NE. For a policy $\pi^{\mathcal{I}}$ and its induced distribution flow $\mu^{\mathcal{I}}$, the exploitability for the $\lambda$-regularized GMFG is defined as

$$\text{Exploit}(\pi^{\mathcal{I}}) = \int_0^1 \max_{\tilde{\pi} \in \Pi^H} J^{\lambda,\alpha}(\tilde{\pi}, \mu^{\mathcal{I}}) - J^{\lambda,\alpha}(\pi^\alpha, \mu^{\mathcal{I}})\mathrm{d}\alpha.$$

First, the experimental results demonstrate the necessity of modelling the heterogeneity of agents. Figure 1 demonstrates the performance degradation of approximating GMFG by MFG. Here we let the agents play in the GMFG with constant graphons $W_h(\alpha, \beta) = p$ for $p \in \{0, 0.5, 1\}$. The agents have oracle access to the action-value function. We observe that this approximation results in gross errors for learning the NEs of GMFGs with non-constant graphons.

Second, the experiments show that the algorithms designed for unregularized GMFG cannot learn the NE of regularized GMFG. We implement the discrete-time version of the algorithm in Fabian et al.

[2022]; results marked "Unreg PMD" show that the exploitability first decreases and then increases. In line with the discussion in Section 5.2, this originates from keeping too much gradient knowledge in previous iterates $\pi_t^{\mathcal{I}}$. The gradient of the policy is largely correct in the several initial iterations, but a large amount of past knowledge results in it deviating in later iterations, since the past knowledge accumulates. In contrast, our algorithm *discounts* the past knowledge as $(\pi_t^{\mathcal{I}})^{1-\lambda\eta_t}$.

Finally, the results indicate the influence of action-value function estimation. In the experiments, we run our algorithm when $N = 5, K = 300$, $N = 10, K = 100$, and $N = 10, K = 300$. Figure 1 shows that the algorithm with $N = 10, K = 300$ can achieve a smaller error than the algorithms both with $N = 10, K = 100$ and $N = 5, K = 300$. This is in agreement with Theorem 4.

## 7 Conclusion

In this paper, we focused on two fundamental problems of $\lambda$-regularized GMFG. Firstly, we established the existence of NE. This result greatly weakened the conditions in the previous works. Secondly, the provably efficient NE learning algorithms were proposed and analyzed in the weakly monotone GMFG motivated by Lasry and Lions [2007]. The convergence rate of MonoGMFG-PMD features the first performance guarantee of discrete-time algorithm without extra conditions in monotone GMFGs. We leave the lower bound of this problem to the future works.

## Acknowledgments and Disclosure of Funding

Fengzhuo Zhang and Vincent Tan acknowledge funding by the Singapore Data Science Consortium (SDSC) Dissertation Research Fellowship, the Singapore Ministry of Education Academic Research Fund (AcRF) Tier 2 under grant number A-8000423-00-00, and AcRF Tier 1 under grant numbers A-8000980-00-00 and A-8000189-01-00. Zhaoran Wang acknowledges National Science Foundation (Awards 2048075, 2008827, 2015568, 1934931), Simons Institute (Theory of Reinforcement Learning), Amazon, J.P. Morgan, and Two Sigma for their support.

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
