# OpenReview forum: "Learning Regularized Monotone Graphon Mean-Field Games"
_NeurIPS.cc/2023/Conference — NeurIPS 2023 poster_

### Official Review · Reviewer_NT3m · 2023-06-27

**Soundness:** 3 good
**Presentation:** 3 good
**Contribution:** 3 good
**Rating:** 6
**Confidence:** 2

**Summary:**

The paper focuses on two fundamental problems in regularized Graphon Mean-Field Games (GMFGs). The first problem is to establish the existence of a Nash Equilibrium (NE) of any $\lambda$-regularized GMFG (for $\lambda \geq 0$). The second problem is to propose provably efficient algorithms to learn the NE in weakly monotone GMFGs. Regarding the first problem,  this paper used weaker conditions than previous works analyzing unregularized GMFGs ($\lambda = 0$) or $\lambda$-regularized MFGs, which are special cases of $\lambda$-regularized GMFGs. To address the second problem, the paper proposes a discrete-time algorithm and derives its convergence rate solely under weakly monotone conditions. Furthermore, the paper develops and analyzes the action-value function estimation procedure during the online learning process, which is absent from algorithms for monotone GMFGs. The efficiency of the designed algorithm is corroborated by empirical evaluations.


**Strengths:**

1. It's quite fascinating to uncover the link between MFG and GMFG as explored in the paper. Notably, the use of lambda-regularized MFG in Proof 1 to find the NE of regularized GMFG provides an intuitive understanding.

2. PMD for the general function approximation is impressive. The decision to employ policy mirror descent adds an interesting dimension to the methodology.

The paper appears to be well-rounded and articulately composed. It has a clear presentation of its findings.

**Weaknesses:**

The paper does have quite a few notations, but it's understandable considering the complexity of GMFG. I've got a few questions, which might also point out some areas in the paper that could be improved. I've put these questions and potential weaknesses together in the question section for easy reference.

**Questions:**

I mainly work on Stochastic game theory, not mean field game, so some of my questions might be quite basic.

1. From the first strength, I'm curious about how an NE of the $\lambda$-regularized GMFG is built from an NE of the made-up $\lambda$-regularized MFG. What makes this possible? This is really interesting to me. Is this a common method for MFG and GMFG? Also, is step 3 in the proof of theorem 1, something new compared to the previous literature?

2. In stochastic game theory, I know that $\lambda$-regularizing usually gives a better landscape for optimization, making it easier to find (and define) NE in the regularized setup. Is it the same in the MFG setting?

3. Is the analysis linked to this paper? It'd be helpful to know the technical differences between [1] and this paper.
[1] Zhan, Wenhao, et al. "Policy mirror descent for regularized reinforcement learning: A generalized framework with linear convergence." SIAM Journal on Optimization 33.2 (2023): 1061-1091.

If this is well-addressed, I am down to re-evaluate this paper.

**Limitations:**

The authors adequately addressed the limitation and potential negative societal impact on their work.

---

> ### Author Rebuttal · Authors · 2023-08-09
>
> We thank the reviewer for the valuable feedback. We address the major concerns in the following.
>
> **Relationship between NEs of $\lambda$-regularized GMFGs and MFGs**
>
> To build an NE of $\lambda$-regularized GMFGs from an NE of the constructed $\lambda$-regularized MFG, we take the position of the agent as a state in the MFG. This method is partially covered by [1], but we are the first to provide the mathematically strict proof of this method.
>
> **Novelty of Step 3 in the Proof of Theorem 1**
>
> Yes, the work  [1] only considers the unregularized case in Step 3. In contrast, we cope with the regularized MFGs and prove Theorem 2. This requires us to prove the corresponding regularized game operator is closed, and we achieve this by constructing a set $A_{h}^{(n)}$ in Lines 721 and 722 and proving Proposition 8.
>
> **Regularization Explanation**
>
> Regularization is a standard technique in game theory. In GMFGs, it helps us to uniquely define the optimal policy of the MDP induced by any distribution flow. The regularization also makes sure that the optimal policy is not a degenerate distribution, and thus makes KL divergence an appropriate performance metric for us to analyze.
>
>
> **Comparison With Zhan et al. 2023**
>
> We would like to highlight that we consider a different problem as [2]. They analyze the performance of policy mirror descent in a fixed MDP. In contrast, we analyze the policy mirror descent in a non-stationary MDP, and the non-stationarity originates from that the reward functions change according to the distribution flows induced by the policies.
>
> [1] K. Cui, and H. Koeppl. Learning graphon mean field games and approximate Nash equilibria. In International Conference on Learning Representations (2022).
>
> [2] Zhan, Wenhao, et al. Policy mirror descent for regularized reinforcement learning: A generalized framework with linear convergence. SIAM Journal on Optimization 33.2 (2023): 1061-1091

---

> > ### Comment · Reviewer_NT3m · 2023-08-10
> > **Thank you**
> >
> > Thank you very much!

---

### Official Review · Reviewer_veSy · 2023-07-06

**Soundness:** 3 good
**Presentation:** 3 good
**Contribution:** 3 good
**Rating:** 7
**Confidence:** 3

**Summary:**

The paper analyzes policy mirror descent for solving regularized GMFG. The results provide new guarantees for learning GMFG without stringent oracle assumptions, and unlike some past works, it does not restrict the results to continuous time analysis. Furthermore, the paper provides an analysis of the case of function approximation. Experimental results are also presented for certain GMFG.

**Strengths:**

The paper provides convincing theoretical guarantees for the proposed algorithm. Table 1 is in general convincing of the theoretical contributions of the work. The analysed setting is novel and removes theoretical oracle assumptions in past work. Furthermore, the analysis is in discrete time (i.e., purely algorithmic) and not in continuous time dynamics.

The theoretical results are very clearly presented, and the assumptions are explicit. There is no ambiguity, and the proofs seem correct (although I might have missed details).

Furthermore, the incorporation of a function class in the analysis as opposed to oracle access is new in MFG/GMFG to the best of my knowledge.

**Weaknesses:**

While the provided result for offline RL-based approximation of value functions is interesting theoretically, it might be prohibitive in practice as the results stand.

Table 1 seems to indicate a large variety of assumptions employed in literature. It is not directly clear how the setting compares to other alternative settings, for instance, if related, a comparison of weak monotonicity with other definitions of monotonicity as well as contraction. This makes it difficult to compare the theory.

Experimental results are restricted to toy problems, however, it is possible that no alternative benchmarks exist for GMFG.

**Questions:**

How does the function class complexity $\mathcal{N}_\infty$ scale in general with various schemes of approximation? Several examples could illuminate the reader. Similarly, making the definition of the covering number in the main body explicit could help reading easier.

Line 230: The statement regarding switching policies is not clear to me. In general, is there an intuitive explanation of the weak monotonicity assumption?

How do the results compare to the specific case of MFG? The graphon structure seems to be more general than MFG, admitting it as a special GMFG with a particular graphon. For a clear comparison, it could be interesting to state the implied bounds (if any) for the special case of monotone MFG with the corresponding implication made clear. Otherwise a direct comparison might be difficult with continuous time results in MFG.


**Limitations:**

Potential additional limitations to be discussed were mentioned in the weaknesses and limitations.

---

> ### Author Rebuttal · Authors · 2023-08-09
>
> We thank the reviewer for the valuable feedback. We address the major concerns in the following.
>
> **Empirical Action-Value Function Estimation and Simulation Benchmark**
>
> Our work focuses on the optimization complexity and the sample complexity of the algorithms. The efficacy of our proposed algorithms, including the action-value function estimation, is corroborated in the simulation results. We note that there is no benchmark for GMFGs, and we follow the experiment settings in the previous works [1,2].
>
> We note that Line 7 in Algorithm 2 may incur potentially computational cost for the action-value function estimation. However, this step has closed-form expressions for the tabular case, which includes a wide range of applications[3]. For the more complex environment, this step can be implemented via gradient descent. We leave the application of our algorithms, especially the action-value function estimation, on them as future work.
>
> **Comparison of Different Monotone Conditions and Contraction**
>
> We first provide a comparison between the different monotone conditions. Our work, [1], and [4] all consider the monotone condition for multi-population/graphon mean-field games. [1] defines the inequality in our Proposition 2 as the monotone condition. [4] defines the monotone condition for multi-population MFGs, and the monotone condition in our work can recover their definition by setting the graphons as block-wise constant graphons in Definition 3. [2] and [5] consider the monotone condition for MFGs. The definition of monotone condition in [5] is a special case of the definition in [4] by setting the number of populations as one. The definition in [2] is a special case of [5], where the reward functions are the sum of the distribution flow-independent part and the distribution flow-dependent part.
>
> The literature of MFGs, a subclass of GMFGs, is split into two threads: the contraction condition and the monotone condition. We note that the comparison between the monotonicity and the contraction is open even for MFGs.
>
>
> **Example of Covering Numbers**
>
> We give some examples of the covering number of the function class. Consider a one-dimensional parametric function class $\mathcal{F} _{\mathrm{exp}}=\lbrace f _{\theta}:[0,1]\rightarrow\mathbb{R}\,|\,\theta\in[0,1]\rbrace$, where $f _{\theta}= 1-\exp(\theta x)$. The covering number is $\log \mathcal{N} _{\infty}(\delta,\mathcal{F} _{\mathrm{exp}})\asymp \log(1/\delta)$ as $\delta\rightarrow 0$ [6]. Consider a non-parametric function class $\mathcal{F} _{L}=\lbrace g:[0,1]\rightarrow\mathbb{R}\,|\, g(0)=0, |g(x)-g(y)|\leq L|x-y| \text{ for all }x,y\in[0,1]\rbrace$. The covering number is $\log \mathcal{N} _{\infty}(\delta,\mathcal{F} _{\mathrm{exp}})\asymp L/\delta$ as $\delta\rightarrow 0$ [6].
>
>
> **Explanation of the Switching Policies Argument**
>
> The statement regarding switching policies can be clarified through Proposition 2. Here we note that Proposition 2 states that for $\pi^{\mathcal{I}}$, $\tilde{\pi}^{\mathcal{I}}$ and the corresponding distribution flows $\mu^{\mathcal{I}}$,$\tilde{\mu}^{\mathcal{I}}$, implementing policies $\tilde{\pi}^{\mathcal{I}}$, $\pi^{\mathcal{I}}$ on the MDPs induced by $\mu^{\mathcal{I}}$, $\tilde{\mu}^{\mathcal{I}}$ will have higher rewards than implementing policies $\tilde{\pi}^{\mathcal{I}}$, $\pi^{\mathcal{I}}$ on the MDPs induced by themselves. Thus, switching policies can get higher rewards on at least one of the MDPs induced by $\mu^{\mathcal{I}}$ and $\tilde{\mu}^{\mathcal{I}}$.
>
> **Comparision with MFG results**
>
> We highlight that our results for GMFGs directly imply the results for MFGs when the graphons are constant graphons. For example, with constant graphons, Theorem 3 implies that the convergence rate of Algorithm 1 for MFGs is $\tilde{O}(T^{-1/2})$ in the sense of the KL divergence. Theorem 3 in [4] shows that with the additional potential structure, the fictitious play algorithm converges at rate $O(T^{-1/2})$ in the sense of exploitability. We can prove that the performance guarantee in KL divergence implies that in exploitability, and the resultant bound for our algorithm is $\mathrm{Exploit}(\hat{\pi})=\tilde{O}(T^{-1/4})$. Although the rate is slower that $O(T^{-1/2})$ in [4], we do not require the potential structure. The reason for an additional square root here is that we adopt Pinsker’s inequality, and this may be improved with a tighter distribution flow error analysis than the following analysis.
>
> Here we specify the main proof ideas. For a policy $\pi$ that is close to NE policy $\pi^{\ast}$ in KL, we denote the distribution flow induced by $\pi$ as $\mu$ and denote the optimal policy on the MDP induced by $\mu$ as $\tilde{\pi}^{\ast}$. We aim to prove that the difference between the cumulative rewards of $\tilde{\pi}^{\ast}$ and $\pi$ on $\mu$ is bounded. This can be achieved by noting that: 1. The performance difference lemma can bound this with the Total Variation (TV) between $\pi$ and  $\tilde{\pi}^{\ast}$. 2. The TV between $\tilde{\pi}^{\ast}$ and $\pi^{\ast}$ is proportional to the TV between $\mu$ and $\mu^{\ast}$, since they are the optimal policies on the MDPs induced by $\mu$ and $\mu^{\ast}$. 3. $\mu$ and $\mu^{\ast}$ are close since they are induced by $\pi$ and $\pi^{\ast}$.
>
> [1] C. Fabian, K. Cui, and H. Koeppl. Learning sparse graphon mean field games. Aistats PMLR, 2023.
>
> [2] S. Perrin, et al. Fictitious play for mean field games: Continuous time analysis and applications. Neurips 33 (2020).
>
> [3] A. Aurell, et al. "Finite state graphon games with applications to epidemics." Dynamic Games and Applications 12.1 (2022).
>
> [4] J. Perolat, et al. Scaling up mean field games with online mirror descent. arXiv:2103.00623(2021).
>
> [5] M. Geist, et al. Concave utility reinforcement learning: the mean-field game viewpoint. arXiv:2106.03787(2021).
>
> [6] M. J. Wainwright. High-dimensional statistics: A non-asymptotic viewpoint. Cambridge university press, 2019.

---

> > ### Comment · Reviewer_veSy · 2023-08-14
> >
> > Thank you for the very detailed rebuttals. All of my questions have been answered, and I will leave my rating as a 7.

---

### Official Review · Reviewer_Epba · 2023-07-08

**Soundness:** 4 excellent
**Presentation:** 3 good
**Contribution:** 3 good
**Rating:** 7
**Confidence:** 3

**Summary:**

Intuitively, a "graphon mean field game" (GMFG) describes the large-$N$ limit of a game with $N$ players, where the payoff of a player $i$ depends on a weighted average of the states of other players $j\in [N]$. The graphon aspect comes from the fact that players have "identities" given by numbers $U_i\in [0,1]$, and the weights in the averages are of the form $W(U_i,U_j)/N$, where $W$ is a suitable function (if $W$ is constant, we have a regular mean field game MFG). GMFGs are potentially useful in multiagent reinforcement learning whenever agent interactions are not too strong.

The present paper does not deal with the finite-$N$ problem, but rather with its continuous limit. One important point is that it considers regularized versions of GMFGs, with an added penalization term. The main results are as follows.

* Theorem 1 is a new result on the existence of equilibria for (potentially regularized) GMFGs. The main attraction of this result, in comparison with previous work, is that it only makes continuity assumptions on the reward function and transition probabilities (the function $W$ is still assumed Lipschitz). Moreover, the regularization had not been considered previously. Theorem 1 is obtained via a careful reduction to a non-graphon Mean Field Game, for which the authors also prove a new existence result (Theorem 2).

* The paper then considers algorithms for approximating Nash equilibria for GMFGs that satisfy a monotonicity condition. Theorem 3 obtains a result under the existence of an "action function oracle". When that is not available, one can resort to function approximation: this leads to Theorem 4, which works under additional assumptions (and require the regularization).

A small set of experiments suggests that the authors' method performs well in practice, and also that regularization is important to achieve good performance.



**Strengths:**

The paper is original and significant. It is also quite clear. The following two points stand out.

* The existence results work under weak conditions because of the clever use of "soft" techniques.
* The algorithms work under relatively natural conditions.

**Weaknesses:**

* It is not clear to me if the Lipschitz assumption on $W$ is really needed.
* The proof of Theorem 4 is not particularly surprising. (I hesitate to call this a "weakness", but it is true that this part of the analysis is not too surprising.)
* In certain settings the user may be interested in the unregularized GMFG; however, Theorem 4 requires nonzero regularization. The paper does not provide bounds on how close a (near-)NE for the regularized case is to being a NE for the unregularized game.



**Questions:**

These are simply rewordings of my comments on weaknesses.

Q1: Regarding Theorems 1 and 2, it looks like the only place where the Lipschitz assumption on $W$ is used is (D.5). However, it looks that all that is needed is uniform continuity, which is equivalent to continuity.

Q2: In certain settings the user may be interested in the unregularized GMFG; however, Theorem 4 requires nonzero regularization. Can one prove bounds on how close a (near-)NE for the regularized case is to being a NE for the unregularized game?

**Limitations:**

None were discussed, but I don't think there was any need to do so.

---

> ### Author Rebuttal · Authors · 2023-08-09
>
> We thank the reviewer for the valuable feedback. We address the major concerns in the following.
>
> **Lipschitz Continuity of $W$**
>
> We thank the reviewer for this suggestion. For the proof of Theorem 1, the Liphschitz assumption on $W$ is only used to establish (D.5), and this can be proved with continuity assumption. We will modify this in the revised version.
>
> **Novelty of Theorem 4**
>
> The learning error bound in Theorem 4 mainly consists of two terms: the optimization error and the estimation error. We would like to discuss them separately.
>
> First, the optimization error analysis in Theorem 4 is a direct adaption of Theorem 3. This is the first analysis of the discrete-time algorithm only under the monotone condition even in the Mean-Field Games (MFG) problem, a subclass of Graphon Mean-Field Games (GMFG) with constant graphons. Here we make use of the intuition provided by Proposition 2 to guide the analysis of the policy mirror descent in a non-stationary environment.
>
> Second, the estimation error is analyzed with the general function approximation and provides the error dependency on the number of episodes $K$ and the number of sample agents $N$. This analysis is important for the realistic application of the algorithm, where the algorithm needs to estimate the action-value function from samples. Our results provide the guidance to choose the parameters $T$, $N$, and $K$ to achieve a learning error $\varepsilon$.
>
> **Regularization Explanation**
>
> First, we note that the regularized setting itself is important in the MFGs, a subclass of GMFGs. In the real-world applications, there are usually constraints about the safety and the incentives for the exploration in the learning process, and people formulate these requirements via regularizations. The regularized setting is widely studied in MFGs [1,2,3]. Our work follows this thread and consider learning NE for the regularized GMFGs.
>
> Second, we would like to quantify the difference between the NEs of the regularized GMFGs and the unregularized GMFGs. Here, we denote the NE of the $\lambda$-regularized one as $(\pi^{\ast,\mathcal{I}},\mu^{\ast,\mathcal{I}})$ and denote the optimal policy of the unregularized MDP induced by $\mu^{\ast,\mathcal{I}}$ as $\pi^{\mathcal{I}}$. Now we examine how far $(\pi^{\ast,\mathcal{I}},\mu^{\ast,\mathcal{I}})$ is from the NE of the unregularized GMFGs. We note that $\mu^{\ast,\mathcal{I}}$ is induced by the policy $\pi^{\ast,\mathcal{I}}$, and thus they satisfy the distribution consistency condition. For the player rationality condition, Theorem 2 in [4] shows that $V_{1}^{0,\alpha}(s,\pi^{\alpha},\mu^{\ast,\mathcal{I}})\leq V_{1}^{\lambda,\alpha}(s,\pi^{\alpha},\mu^{\ast,\mathcal{I}})+\lambda H\log|\mathcal{A}|$ for all $\alpha\in[0,1]$ and $s\in\mathcal{S}$. Therefore, $(\pi^{\ast,\mathcal{I}},\mu^{\ast,\mathcal{I}})$ satisfies the player rationality condition up to $\lambda H\log|\mathcal{A}|$. We highlight that this gap also appears in the MFG works [1,2]. Our work focuses on learning the regularized GMFGs and leaves easing this gap as the future work.
>
> [1] B. Anahtarci, C. D. Kariksiz, and N. Saldi. Q-learning in regularized mean-field games. Dynamic Games and Applications 13.1: 89-117  (2023).
>
> [2] Q. Xie, Z. Yang, Z. Wang, and A. Minca. Learning while playing in mean-field games: Convergence and optimality. In International Conference on Machine Learning (pp. 11436-11447). PMLR (2021).
>
> [3] B. Yardim, S. Cayci, M. Geist, N. He. Policy mirror ascent for efficient and independent learning in mean field games. In International Conference on Machine Learning (pp. 39722-39754). PMLR (2023).
>
> [4] M. Geist, B. Scherrer, and O. Pietquin. A theory of regularized markov decision processes. International Conference on Machine Learning. PMLR, (2019).

---

> > ### Comment · Reviewer_Epba · 2023-08-14
> > **Thank you**
> >
> > I thank the authors for their rebuttal. I remain favorable to the paper being accepted.

---

### Official Review · Reviewer_eGqz · 2023-07-25

**Soundness:** 4 excellent
**Presentation:** 4 excellent
**Contribution:** 2 fair
**Rating:** 6
**Confidence:** 3

**Summary:**

This paper studies regularized Graphon Mean-Field Games (GMFGs). They make two theoretical improvements over previous works on this topic:
* They prove existence of Nash equilibrium under weaker assumptions (e.g., weaker requirement on the continuity of the game) than previous works.
* For the special case of monotone regularized GMFGs, they give an mirror-descent algorithm that learns the unique Nash equilibrium. Compared to previous works, the novelty here is that the algorithm works for regularized games in discrete time (as opposed to unregularized games and continuous time).

In terms of techniques, their first result follows the proof plan of Cui and Koeppl, [2021] that reduces the problem to proving existence of Nash equilibrium for a subclass of GMFGs called MFGs. Their main technical contribution is proving an equilibrium existence result (Theorem 2) for MFGs under weaker assumptions using a different approach than previous works. Their second result essentially adapts the algorithm from Perolat et al. [2021] to their setting.

**Strengths:**

This paper builds upon previous works and makes reasonable improvements. Most interestingly, the condition of their equilibrium existence result (Theorem 2) seems to be significantly weaker than previous works. The paper is well-written. They did a great job introducing the problem and the results and explaining the techniques and the difference from previous works.

**Weaknesses:**


As someone who is not closely following this line of work, it is hard for me to gauge the significance of the new equilibrium existence result, i.e., whether the weakened assumption is significantly more applicable than the assumptions made in previous works. The paper briefly mentions the assumptions in previous works are ``overly restrictive for real-world applications'' but did not provide any concrete example.

The algorithm for learning Nash equilibrium in their setting seems to be rather straightforward adaptation (with more careful analysis) from previous work of Perolat et al. [2021].

**Questions:**

Could you elaborate why the new equilibrium existence results are more applicable than previous works with concrete examples?

What are the concrete benefits of the exploration step (Line 5 of Algorithm 1) in your learning algorithm? Is it necessary for proving your result?

---

> ### Author Rebuttal · Authors · 2023-08-09
>
> We thank the reviewer for the valuable feedback. We address the major concerns in the following.
>
> **Detailed Comparison of NE Existence Conditions**
>
> Theorems 1 and 2 in our work,  Proposition 3 in [1], and Proposition 3 in [2] all derive the existence of Nash Equilibrium (NE) in the regularized Mean-Field Games (MFG) or Graphon Mean-Field Games (GMFG). Our results in Theorems 1 and 2 hold for continuous reward functions, weakly continuous transition kernel, Lipschitz graphons, and any value of regularization parameter $\lambda\geq 0$. In contrast, Proposition 3 in [1] proves the existence of NE via contraction, which requires that $K_{H}<1$ therein. From the definition of $K_{H_1}$ and $K_{1}$, the condition $K_{H}<1$ requires that the Lipschitz constants of transition kernels, and reward functions to be sufficiently small. Proposition 3 in [2] requires that the regularization parameter $\lambda$ should be sufficiently larger than the Lispchitz constants of the game operators. These conditions are restrictive for real-world applications. In addition, our results hold for compact state space, but [2] is constrained to the setting with finite state space.
>
> Theorem 1 in our work, Theorem 1 in [3], and Theorem 1 in [4] all consider the existence of NE in GMFGs. However, [3] and [4] only prove the existence of NE in the unregularized case under the Lipschitz continuity of the transition kernels and reward functions. In contrast, our work provides the results for all $\lambda\geq 0$, including both the regularized and unregularized cases, under the continuity of reward functions and the weak continuity of transition kernels.
>
> **Algorithm Comparison With Perolat et al. [2021]**
>
> We would like to highlight three main differences between our algorithm and the algorithm in [5].
>
> First, our provably efficient algorithm is discrete-time, while the one in [5] is continuous-time. In the realistic application of the algorithm, the optimization can only be implemented in a discrete-time manner, which introduces additional quantization error for the one in [5]. In contrast, our discrete-time algorithm does not suffer from this additional error.
>
> Second, our optimization algorithm does not assume that we have access to the nominal value of the action-value functions, while [5] only analyzes the algorithm with the true action-value functions. The analysis of the algorithm with action-value function estimates requires a perturbation analysis of the algorithm, which is important for applications to realistic scenarios.
>
> Finally, our algorithm has an additional exploration procedure that is not included in [5]. Intuitively, this procedure guarantees that the support of the NE policy is contained in that of the policy estimate in each step, which accommodates the potential error originating from the action-value function estimation and the quantization. Technically, this guarantees that the KL divergence between the NE policy and the policy estimate in each iteration is finite, as shown in (F.1) in Appendix.
>
> [1] B. Anahtarci, C. D. Kariksiz, and N. Saldi. Q-learning in regularized mean-field games. Dynamic Games and Applications 13.1 (2023): 89-117.
>
> [2] K. Cui, and H. Koeppl. Approximately solving mean field games via entropy-regularized deep reinforcement learning. International Conference on Artificial Intelligence and Statistics. PMLR, 2021.
>
> [3] K. Cui, and H. Koeppl. Learning graphon mean field games and approximate Nash equilibria. In International Conference on Learning Representations (2022).
>
> [4] F. Christian, K. Cui, and H. Koeppl. Learning sparse graphon mean field games. International Conference on Artificial Intelligence and Statistics. PMLR, 2023.
>
> [5] J. Perolat, et al. "Scaling up mean field games with online mirror descent." arXiv preprint arXiv:2103.00623 (2021).

---

> > ### Comment · Reviewer_eGqz · 2023-08-11
> >
> > My question was could you give some concrete examples for your claim `` These conditions are restrictive for real-world applications'' but the conditions of your existence theorem are not restrictive? This is supposed to be the main motivation of this paper, but the paper did not mention any concrete example of such applications. (I understand your existence theorem requires weaker assumptions in many aspects theoretically.)

---

> > > ### Author Response · Authors · 2023-08-14
> > > **Response to Reviewer eGqz (Part 1/2)**
> > >
> > > Thanks for the reply. We would like to give some concrete examples of the existence of NE in MFGs. The results in [1] can be easily generalized to the finite-horizon MDP with undiscounted rewards for comparison. It requires the reward functions and the transition kernels are Lipschitz functions.
> > >
> > > \begin{align*}
> > >     |r _{h}(s _{h},a _{h},\mu _{h}) - r _{h}(s _{h}^{\prime},a _{h}^{\prime},\mu _{h}^{\prime})|&\leq L _1\cdot(\mathbb{I}\lbrace s _h\neq s _h^{\prime}\rbrace +2\mathbb{I}\lbrace a _h\neq a _h^{\prime}\rbrace+\Vert \mu _h-\mu _h^{\prime}\Vert _1)\\\\
> > >     \Vert P _{h}(\cdot| s _{h}, a _{h},\mu _h)-P _{h}(\cdot| s _h^{\prime}, a _h^{\prime},\mu _h^{\prime})\Vert _{1}&\leq K _1\cdot(\mathbb{I}\lbrace s _h\neq s _h^{\prime}\rbrace +2\mathbb{I}\lbrace a _h\neq a _h^{\prime}\rbrace+\Vert\mu _h-\mu _h^{\prime}\Vert _1)
> > > \end{align*}
> > > for all $h\in[H]$. Then we define the $Q _{\mathrm{Lip}}$ as $Q _{\mathrm{Lip}}=L _1(1-(K _1/2)^H)/(1-K _1/2)$. Then the contraction constant in Proposition 3 in [1] for finite-horizon MFG with undiscounted rewards is
> > > \begin{align*}
> > >     C = 3K _1+\frac{5}{\lambda}\bigg(L _1+\frac{K _1 Q _{\mathrm{Lip}}}{2}\bigg).
> > > \end{align*}
> > > The parameters $K _1,L _1, \lambda$ should guarantee that $C<1$, i.e., $K _1<1/3$, and $\lambda>5(L _1+ K _1Q _{\mathrm{Lip}}/2)/(1-3K _1)$.
> > >
> > > * We consider the suspect susceptible–infected–susceptible(SIS) problem in [2], which is a simplified version of the Susceptible-InfectedRemoved (SIR) problem in [3]. The state space contains the susceptible state $S$ and the infected state $I$, i.e., $\mathcal{S}=\lbrace S,I\rbrace$, and the action space contains the going out $U$ and keeping distance $D$, i.e., $\mathcal{A}=\lbrace U,D \rbrace$. The reward function is defined as $r _{h}(s _{h},a _{h},\mu _{h})=-r _{I}\mathbb{I}\lbrace s _h=I\rbrace –r _{D} \mathbb{I}\lbrace a _h=D\rbrace$, and the transition kernel is defined via
> > > \begin{align*}
> > > 	P _{h}(s _{h+1}=I | s _h=S, a _h=U,\mu _h) &= P _{a} \cdot \mu _{h}(I)\\\\
> > > 	P _{h}( s _{h+1}=S| s _h=I,\cdot,\cdot) &= P _{r}\\\\
> > > 	P _{h}( s _{h+1}=I| s _h=I, a _h=D,\cdot) &= 0
> > > \end{align*}
> > > for all $h\in[H-1]$. The first equation indicates that an outside susceptible person will be infected with probability proportional to the ratio of infected people. The parameters of this problem are $r _I,r _D,P _a,P _r>0$. We can show that
> > > \begin{align*}
> > >     |r _{h}(s _{h},a _{h},\mu _{h})-r _{h}(s _{h}^{\prime},a _{h}^{\prime},\mu _{h})^{\prime}|&\leq \max\lbrace r _{I},r _D/2\rbrace\cdot(\mathbb{I}\lbrace s _h\neq s _h^{\prime}\rbrace +2\mathbb{I}\lbrace a _h\neq a _h^{\prime}\rbrace+\Vert\mu _h-\mu _h^{\prime}\Vert _1)\\\\
> > >     \Vert P _{h}(\cdot| s _h, a _h,\mu _h)-P _{h}(\cdot| s _h^{\prime}, a _h^{\prime},\mu _h^{\prime})\Vert _{1}&\leq \max\lbrace 1-P _r,P _a\rbrace\cdot(\mathbb{I}\lbrace s _h\neq s _h^{\prime}\rbrace +2\mathbb{I}\lbrace a _h\neq a _h^{\prime}\rbrace+\Vert\mu _h-\mu _h^{\prime}\Vert _1)
> > > \end{align*}
> > > To guarantee that $C<1$, we require that $P _r>2/3, P _a<1/3$ and $\lambda$ is larger enough than the rewards $r _{I}, r _D$, which restricts the regularization, the infection and recovery probability of each agent. The results in our work do not impose this constraint.
> > >
> > > * We consider the linear quadratic MFG in [4]. The state space is $\mathcal{S}=\lbrace -L,\cdots,L\rbrace$, and the action space is $\mathcal{A}=\lbrace -M,\cdots,M\rbrace$. The reward function and the transition kernel are defined as
> > > \begin{align*}
> > >     r _{h}(s _{h},a _{h},\mu _{h})&= -\frac{1}{2}a _h^2+qa _h(m _h-s _h)-\frac{\kappa}{2}(m _h-s _h)^2\\\\
> > >     s _{h+1}&= \mathrm{Discretize}[s _h+ a _h + K(m _h-s _h)+\sigma\varepsilon _h]
> > > \end{align*}
> > > for $h\in[H]$, where $q,\kappa,K>0$ are the parameters of the game, $m _h=\sum _{s\in\mathcal{S}}s\cdot\mu _h(s)$ is the first moment of the state distribution, $\varepsilon\sim\mathcal{N}(0,1)$ is the noise, and $\mathrm{Discretize}[\cdot]$ operator discretizes a state into the closed state in $\mathcal{S}$. With the data processing inequality and some basic calculations, we can show that
> > > \begin{align*}
> > >     |r _{h}(s _{h},a _{h},\mu _{h})-r _{h}(s _{h}^{\prime},a _{h}^{\prime},\mu _{h})^{\prime}|&\leq \max\lbrace (M+2qL)^2/4,L(3\kappa L+2qM)\rbrace\cdot(\mathbb{I}\lbrace s _h\neq s _h^{\prime}\rbrace +2\mathbb{I}\lbrace a _h\neq a _h^{\prime}\rbrace+\Vert\mu _h-\mu _h^{\prime}\Vert _1)\\\\
> > >     \Vert P _{h}(\cdot| s _h, a _h,\mu _h)-P _{h}(\cdot| s _h^{\prime}, a _h^{\prime},\mu _h^{\prime})\Vert _{1}&\leq \frac{\max\lbrace |1-K|,1/2,2LK\rbrace}{2\sigma}\cdot(\mathbb{I}\lbrace s _h\neq s _h^{\prime}\rbrace + 2\mathbb{I}\lbrace a _h\neq a _h^{\prime}\rbrace+\Vert\mu _h-\mu _h^{\prime}\Vert _1).
> > > \end{align*}

---

> > > > ### Author Response · Authors · 2023-08-14
> > > > **Response to Reviewer eGqz (Part 2/2)**
> > > >
> > > > To guarantee that $K _1<1/3$, we require that $2\sigma>3\max\lbrace |1-K|,1/2,2LK\rbrace$. This is intuitive, since when $\sigma$ is much larger than $L$, the noise will be similar to the uniform distribution on $\mathcal{S}$, which makes the state transition unrelated to $s _h,a _h$, and $\mu _h$. We note that the maximal reward an agent can get in each time step is $M^2/2+2qML+2\kappa L^2$. The fulfillment of $\lambda>5(L _1+ K _1Q _{\mathrm{Lip}}/2)/(1-3K _1)$ requires that $\lambda$ is comparable to the maximal reward in each step. In contrast, our results do not require such a constraint.
> > > >
> > > > [1] B. Anahtarci, C. D. Kariksiz, N. Saldi. Q-learning in regularized mean-field games. Dynamic Games and Applications, 13(1), 89-117(2023).
> > > >
> > > > [2] Kai Cui, and Heinz Koeppl. Learning graphon mean field games and approximate nash equilibria. arXiv preprint arXiv:2112.01280 (2021).
> > > >
> > > > [3] A. Aurell, et al. Finite state graphon games with applications to epidemics. Dynamic Games and Applications 12.1 (2022): 49-81.
> > > >
> > > > [4] S. Perrin, J. Pérolat, M. Laurière, M. Geist, R. Elie, O. Pietquin. Fictitious play for mean field games: Continuous time analysis and applications. Advances in Neural Information Processing Systems, 33, 13199-13213(2020).

---

### Decision · Program_Chairs · 2023-09-21

**Decision:**

Accept (poster)

**Comment:**

This paper presents a compelling exploration of regularized Graphon Mean-Field Games (GMFGs) and offers valuable contributions to this research area. It tackles the challenging problems of establishing Nash Equilibrium (NE) existence in regularized GMFGs and devising efficient algorithms for learning NE in weakly monotone GMFGs. Notably, the paper establishes a novel link between regularized GMFGs and regularized Mean-Field Games (MFGs), shedding light on a method for constructing NEs using MFGs as an intermediate step. The provided empirical evaluations, though limited to toy problems, reinforce the practical applicability of the proposed algorithms. While the paper could benefit from improved clarity in some sections and deeper comparisons with related work, it contributes to the understanding and efficient solution of regularized GMFG and constitutes a technically solid contribution.